# Ideas and perspectives: Heat stress: more than hot air

Hans J. De Boeck[1]*, Helena Van De Velde[1,2]*, Toon De Groote[1], Ivan Nijs[1]

[1] Centre of Excellence PLECO (Plant and Vegetation Ecology), Department of Biology, Universiteit Antwerpen (Campus Drie Eiken), Universiteitsplein 1, B-2610 Wilrijk, Belgium

[2] Terrestrial Ecology Unit, Department of Biology, Universiteit Gent, K. L. Ledeganckstraat 35, B-9000 Ghent, Belgium

*equal contribution

*Correspondence to*: Hans J. De Boeck (hans.deboeck@uantwerp.be)

**Abstract.** Climate models project an important increase in the frequency and intensity of heat waves. In gauging the impact on plant responses, much of the focus has been on air temperatures while a critical analysis of leaf temperatures during heat extremes has not been made. Nevertheless, direct physiological consequences from heat depend primarily on leaf rather than on air temperatures. We discuss how the interplay between various environmental variables and the plants' stomatal response affects leaf temperatures and the potential for heat stress by making use of both an energy balance model and field data. The results demonstrate that this interplay between plants and environment can cause leaf temperature to vary substantially at the same air temperature. In general, leaves tended to heat up when radiation was high and when stomates were closed, as expected. But perhaps counterintuitively, also high air humidity raised leaf temperatures, while humid conditions are typically regarded as benign with respect to plant survival since they limit water loss. High wind speeds brought the leaf temperature closer to the air temperature, which can imply either cooling or warming (i.e. abating or reinforcing heat stress) depending on other prevailing conditions. The results thus indicate that heat waves characterized by similar extreme air temperatures may pose little danger under some atmospheric conditions, but could be lethal in other cases. The trends illustrated here should give ecologists and agronomists a more informed indication about which circumstances are most conductive for heat stress to occur.

## 1 Introduction

Current climate change has made heat waves more likely as both the temperature mean and variability are increasing (Schär et al., 2004). Several well-documented heat waves have occurred during the past years such as those of 2003 (Europe), 2010 (Russia) and 2012 (North America), and the likelihood of such major events is expected to increase 5 to 10-fold within the next 40 years (Barriopedro et al., 2011). Heat stress in plants is usually observed when tissue temperatures exceed 40 °C, a threshold that is fairly stable across biomes (Larcher, 2003). Such excessive temperatures affect plant metabolism in multiple ways, ultimately reducing growth and economic yield (Bastos et al., 2014; Chung et al., 2014). This seems at odds with the

reported lack of significant single-factor effects in several ecological studies on heat waves (Poirier et al., 2012; Hoover et al., 2014; De Boeck et al., 2016). We examine here how these seemingly contrasting notions can be reconciled. The fundamental issue is that air temperature ($T_a$) is often considered as an important indicator of heat stress, while metabolic rates and physiological processes are affected much more directly by leaf (tissue) temperatures ($T_l$). Many studies on heat

wave effects do not measure leaf or canopy temperatures and report only on air temperatures (e.g. Bauweraerts et al., 2013; Filewod & Thomas, 2014; Fernando et al., 2014), which suggests an underestimation of the importance of $T_l$ and the variables that influence it. Also in models used to predict heat stress effects, air temperatures are still often used instead of tissue temperatures, as noted by Webber et al. (2016) regarding crop modelling, which can lead to inaccurate predictions of crop yields (Siebert et al., 2014). From literature on environmental biophysics (e.g. Campbell & Norman, 1998; Jones,

2013), we know that leaf and tissue temperatures are determined by a number of environmental conditions (apart from $T_a$, primarily through radiation, wind speed and air humidity) and the stomatal response of the plants. The extent to which these variables can decouple leaf from air temperatures and therefore increase or decrease the potential for heat stress during a heat wave of similar magnitude (in terms of air temperature, as it is usually considered) is discussed here by making use of both an energy balance model based on established physical equations and field data.


## 2 Materials and Methods

The model used to calculate leaf temperature is based on the energy balance equation (Eq. 1):

$$R_{s,in} + R_{l,in} - R_{l,out} - H - \lambda E = 0 \tag{1}$$


The equation states that an equilibrium is reached under a certain set of environmental conditions (the flux of sensible heat H can be either incoming or outgoing), so that the sum of incoming energy (via shortwave radiation $R_{s,in}$ and longwave radiation $R_{l,in}$ absorbed by the leaf) and outgoing energy (outgoing longwave radiation $R_{l,out}$, and latent heat $\lambda E$) is zero. The different terms are derived from other equations, which feature both environmental variables such as wind speed (u) and

relative humidity (RH) of the air, leaf-scale parameters such as stomatal conductance ($g_s$) and characteristic leaf dimension (d), and constants such as the Stefan Boltzman's $\sigma$ ($5.67e^{-8}$ W m$^{-2}$ K$^{-4}$). For more details, we refer to De Boeck et al. (2012). The leaf temperature is calculated in an iterative manner: as a starting situation it is assumed that leaf and air temperature are equal, in which case the energy budget equals zero. In any other situation, the model will assume $T_l$ to be lower/higher than $T_a$ if the energy budget is negative/positive. The iteration proceeds in a stepwise manner, until a precision of 0.01 °C is

achieved. The model was validated earlier (De Boeck et al., 2012), demonstrating a deviation between measured and modelled leaf temperatures of less than 1.5 °C for over 90% of the cases. The model is freely available at <URL>.

In this study, we set $T_a$ at 40 °C to approximate the general threshold for heat stress. Atmospheric pressure (which has limited influence) was kept constant at 100 kPa. Emissivity, reflectivity and absorptivity parameters for leaves and soil were used like in De Boeck et al. (2012). In the main analyses, major inputs, namely incident shortwave energy, stomatal conductance, wind speed, and relative humidity of the air, were varied in a dichotomous manner (high or low) to clearly illustrate the direction of responses. More detailed analyses pairing input variables to better illustrate interrelations are presented as supplementary material. We focus on vegetation represented by species that have narrow leaves (like those found in many grasses) with a characteristic dimension of 0.5 cm, but we also consider the opposite end of the spectrum, namely very broad leaves with a d of 20 cm.

The modelled results are supported by data recorded on five sunny days during a heat wave in Belgium in 2015 (1-5 July). These data were collected at an experimental site in Wilrijk, Belgium on two homogeneous 10 cm tall young grass stands sown five weeks earlier on homogenised soils (Fig. S1). The grass was irrigated daily (c. 5 L m$^{-2}$), with the exception of one day to test the impact of surface drying on the difference between $T_a$ and $T_l$. Radiation sensors (SR03-05, Hukseflux Thermal Sensors, Delft, The Netherlands) had been installed approximately 30 cm above the vegetation, with one sensor directed upwards, and one sensor directed downwards to measure absorbed radiation (the difference between the two readings). At the same height, canopy temperature was recorded with a non-contact thermometer (custom made with a MLX90416ESF sensor, Melexis, Tessenderlo, Belgium). Air temperature and relative humidity were measured at 15 cm height (i.e. just above the canopy) in each plot using custom-made system (with a SHT75 RH/$T_a$ sensor, Sensirion AG, Staefa, Switzerland) shielded from the sun by a thin wooden panel. To ensure that mostly data from times when direct sunshine reached the plots was used (generally between 9 am and 7 pm CET), we omitted data points with absorbed radiation below 100 W m$^{-2}$. This was done to prevent artefacts from dew or times when stomates were still closed.

## 3 Results and discussion

Our results show that high radiation loads are an important prerequisite for heat stress, unless air temperatures exceed the tissue heat stress threshold significantly. Without the energy provided by significant amounts of sunshine, plant tissues will almost always be cooler than the surrounding air, regardless of other conditions (Fig. 1, S2-4). In reality, heat waves usually feature clear skies (De Boeck et al., 2010), implying that high radiation loads during hot weather are probable. This also means that experiments in which high air temperatures are imposed in low-radiation environments, like under laboratory conditions or during overcast days, may underestimate impacts.

As highlighted in earlier studies, water availability or lack thereof is greatly relevant in gauging whether a heat wave will give rise to heat stress (Salvucci and Crafts-Brandner, 2004). If drought prompts a plant to conserve water by lowering stomatal conductance ($g_s$), it warms up as energy dissipation shifts from latent fluxes (providing cooling) to sensible fluxes (increasing temperatures). Because heat and drought often co-occur naturally (De Boeck et al., 2010), this effect is very

relevant in assessing heat wave impacts (Idso, 1982; De Boeck et al., 2016). The potentially misleading nature of $T_a$ in predicting heat stress under varying stomatal conductance is clearly highlighted in our results (Fig. 1, S2, S5-6).

Whenever other conditions alleviate some amount of heat stress (e.g. less radiation, higher $g_s$), more wind would counteract such beneficial effects (Fig. 1, S4-5, S7) through closer coupling between the plant and the air. This may seem counterintuitive as windiness is generally associated with heat dissipation, but the same process also works in the opposite case: when other environmental conditions would exacerbate heat stress, more wind reduces the increase of leaf temperatures. In other words, windy conditions lead to avoidance of the most extreme cases of overheating. Obviously, higher wind speeds promote evapotranspiration, resulting in faster depletion of soil water reserves. This could subsequently lead to lower $g_s$ and thus indirectly promote overheating. As wind speeds in laboratory conditions and/or enclosures are often far below those observed outside (De Boeck et al., 2012), canopy warming may be significantly different from outside as calm conditions tend to exacerbate other effects (Fig. 1, S4-5, S7).

Also for relative air humidity, the results are counterintuitive, with higher humidity more likely to give rise to heat stress (Fig. 1, S2, S6-7). This is caused by slower heat dissipation via transpiration as the water vapour gradient between leaf and air is smaller than in the case of drier air. In fact, the combination of low stomatal conductance and high air humidity causes the greatest warming of leaves above the air temperature (Fig. 1). A five-day period featuring air temperatures at vegetation height exceeding 30 °C every day provided us with an opportunity to test whether increasing air humidity diminishes the cooling capacity of leaves. We indeed found a significant relationship between RH and $T_l - T_a$ (Fig. 2), with $\pm$ 0.84 °C change per 0.1 increase in RH (excluding the dry day). This is comparable to the slope (0.72 °C per 0.1 increase in RH) found with a model run using conditions similar to the heat wave period (Fig. S8). Leaf cooling seemed to be reduced on the only day during which irrigation was withheld (Fig. 2): leaves were warmer than the air 32% of the time on the dry day vs. 4% on days with irrigation (even though incident radiation was c. 15% lower on the dry day, while wind speed was similar). We attribute this relative warming to stomatal closure (leaf wilting observed) resulting from drying of the top soil, and subsequent lower transpiration.

The aforementioned trends were observed both for simulations using narrow (Fig. 1) and also for simulations using bigger leaves (Fig. S9). Any variable increasing the heat load (high radiation) or decreasing heat dissipation (high RH, low wind and $g_s$) led to higher temperature increases in big compared to in small leaves, however. This is no surprise as larger surfaces result in increased decoupling from air temperatures, which can lead to extreme temperature deviations. In cushion plants, which physically act as a giant leaf, increases of tissue temperatures of 20 °C and more above the air temperature have been observed (Gauslaa, 1984), illustrating the importance of physical dimensions in energy balances.

Calculations of leaf temperatures are possible at well-equipped sites applying a model such as the one used here. However, increasing quality and decreasing costs of infrared imaging also enable direct quantification of leaf temperatures and variability thereof. Infrared cameras allow the user to select those pixels or zones deemed most appropriate (e.g. excluding bare soil, focusing only on fully developed leaves), improving control and versatility. Automated measurements and batch image processing can render the entire process more efficient, and allow for a high temporal resolution with limited

workload. Moreover, simultaneous measurements of incoming shortwave radiation enable data filtering (e.g. clear sky, completely overcast), further improving possibilities during data analysis. More technical background information on extrapolation from leaves to canopies, dealing with temperature variability, improving temperature accuracy and automated image recognition can be found in Jones et al. (2009), Jones & Vaughan (2010) and Wang et al. (2010).

In conclusion, we clearly demonstrated that exceedance of critical temperatures in plants depends on more variables than air temperature alone. Radiation, wind speed and relative humidity all affect tissue temperatures, depending on plant water status. This implies that heat waves characterized by the same extreme air temperatures may cause little plant damage under some conditions, but could be detrimental to plant growth and survival in other cases. Although heat stress also depends on other factors, like hardening (Neuner and Buchner, 2012) and development stage (Fischer, 2011), the results from this study can help predict when the probability of heat stress occurring is most likely, and can stimulate ecologists and agronomists to shift the focus beyond merely air temperatures when considering heat waves.

**Author contribution**

H.J. De Boeck and I. Nijs conceptualised the study. T. De Groote developed the model code and H. Van De Velde and H.J. De Boeck performed the simulations. H.J. De Boeck carried out the field measurements. All authors worked together to prepare the manuscript.

**Acknowledgements**

H. Van De Velde was supported by FWO Vlaanderen. We thank F. Kockelbergh for technical assistance and the referees for valuable suggestions.

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

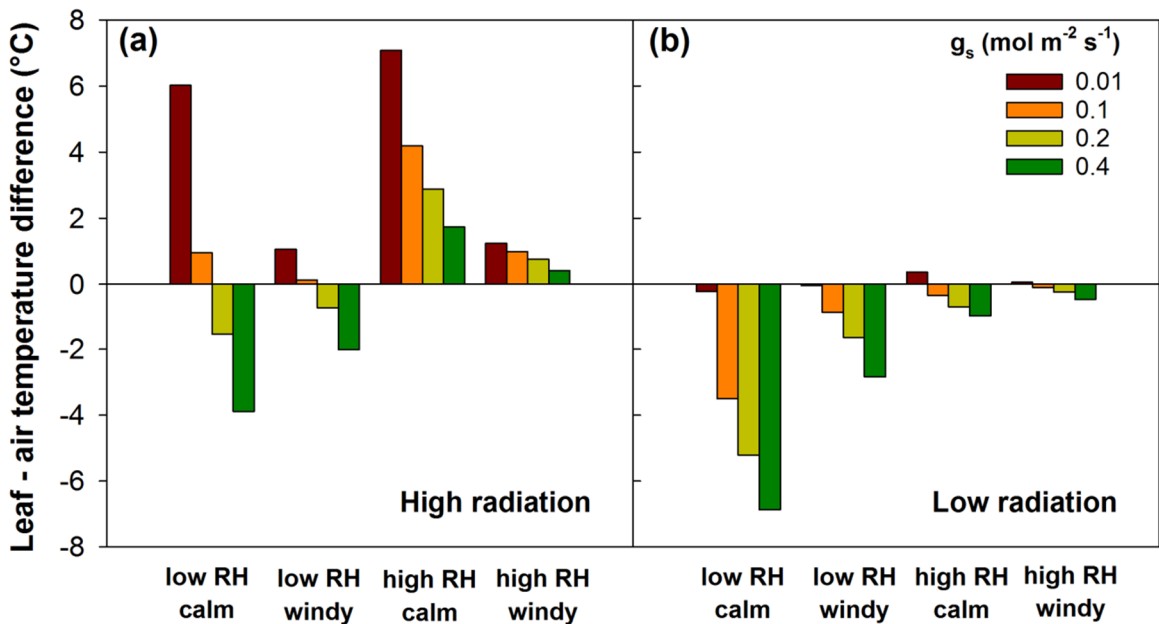

**Figure 1:** Modelled leaf-to-air temperature difference depending on type of heat wave and stomatal conductance ($g_s$). Type of heat wave: high (A) or low (B) incident shortwave radiation (800 or 100 W m$^{-2}$), high or low relative humidity of the air (RH = 0.90 or 0.45), and calm or windy weather (wind speed 0.1 or 6 m s$^{-1}$). Air temperature was set to 40 °C in all simulations, and leaf width to 0.005 m.

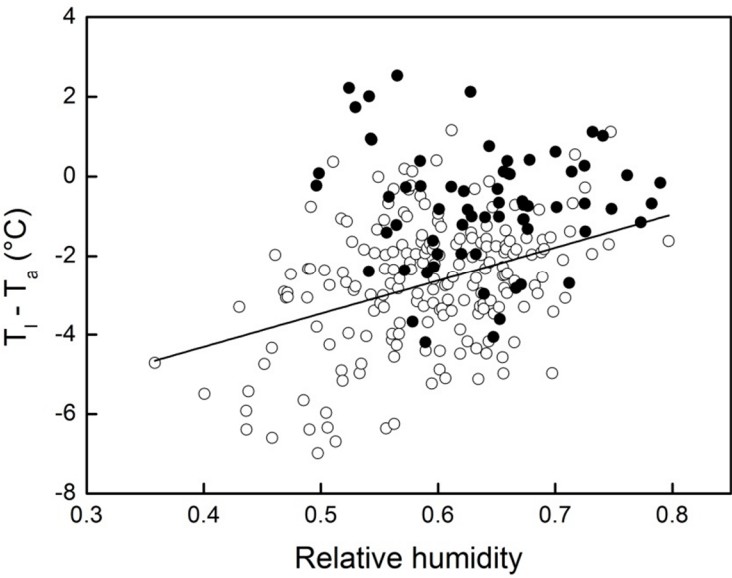

**Figure 2:** Differences between leaf ($T_l$) and air ($T_a$) temperature in function of relative air humidity (RH) measured on a homogeneous grass stand during 5 heat wave days (1-5 July 2015, Belgium). The grass was irrigated daily (white circles), with the exception of one day (black circles). The linear regression (white data points only) was significant at $p < 0.001$ ($R^2 = 0.13$). The difference between regressions (white vs. black) was significant (ANCOVA, $F_{1,257} = 10.3$; $p = 0.001$, Graphpad Prism). In contrast to the model runs, which focus on one peak air temperature (40 °C) to obtain clean comparisons between differing conditions, the relationship presented here contains more scatter because of factors varying throughout the day such as air temperature, incident radiation, stomatal conductance and wind speed.

15