# Peer review of "Figure S1: Modelled leaf-to-air temperature difference depending on type of heat wave and stomatal conductance ( $g_s$ ). Type of heat wave: high (A) or low (B) incident shortwave radiation ( $800$ or $100 \text{ W m}^{-2}$ ), high or low relative humidity of the air ( $\text{RH} = 0.90$ or $0.4"

_Biogeosciences, 2016_

## Referee Comment (RC1) · Anonymous Referee #1 · 28 Apr 2016

"Ideas and perspectives: Heat stress: more than hot air", by De Boeck et al.

De Boeck et al. provide a nice and concise manuscript discussing the importance of the use of leaf temperatures rather than air temperatures for addressing heat stress. The study applies a leaf temperature model published in De Boeck et al. (2012) for a set of sensitivity tests to address the importance of wind speed, relative humidity, radiation levels and leaf size for leaf temperatures. These sensitivities are used to discuss the variations in leaf temperature that can arise through meteorological conditions. In addition, leaf temperature measurements from a young grass stand are analysed to address the importance of drought for leaf temperatures.

The manuscript is nice and short, and although very concise, I consider the setup chosen here (discussion of the importance illustrated with an idealized set of sensitivity

simulations), appropriate to emphasize the authors' opinion. However, the analysis of the leaf temperature measurements that are used to illustrate the drought impact need further attention - these can be used potentially to validate the model and to emphasize the conclusions drawn from the modelling, but in its current form they are not analyzed in great depth, nor quantitatively compared with the modelling. I would recommend to use these to a greater extent (further comments below).

If the measurements can be integrated more in the rest of the manuscript, I expect this manuscript to be an attractive contribution to the discussion on analysis of heat stress and an important message for impact studies.

Major comments:

The data set of leaf temperature is an interesting contribution to the manuscript, but its analysis and the comparison with the modelling is too concise in its current form. I would recommend the following: (1) a statistical analysis of the two sets (with and without irrigation) to determine whether the difference is significant; (2) a validation of the model by using the measured energy fluxes and air temperature to simulate leaf temperature, which can be subsequently compared with the observations (if all model parameters are available or can be estimated); (3) a derivation of the theoretical relationship between the temperature difference and RH using the model, possibly even for different cases (e.g., high/low wind speed), to determine whether the slope found by linear regression reflects the (range of the) theoretical behaviour; (4) clarifying the figure caption of Fig. 2: Is the linear regression for the entire data set, or only for the irrigated days? If the two sets differ significantly, there could be separate regression lines for the two; (5) a discussion on the cause of the high scatter in the observed temperature differences: Are these measurement uncertainties, or can they be explained by the other variables not separated in the figure (stomatal conductance, wind speed).

It is striking to see that there may indeed be a smaller latent cooling for the non-irrigated day (this would need to be confirmed by statistical analysis - the spread is large), but

that there is little impact on the range of observed temperature differences, so the response to other factors that cause the spread may not be affected that much.

Minor comments:

- p. 2, l. 17: the sensible heat flux is not mentioned in the discussion of the components of the energy balance.

- p. 3, l. 7: Please provide more information on the setup with the custom-made non-contact thermometer. I presume it measures infrared radiation? How do you ensure that you measure leaf temperature and not ground temperature? A five-week old and 10 cm high grassland will presumably have a rather low LAI (if LAI was measured, it would be great to have it reported in the manuscript of course).

- p. 3, l. 15: "without extra energy" is somewhat misleading here: This is used to describe the low radiation case (100 W/m2), which of course does resemble a low level of solar energy. A case without extra energy (darkness) would have yet a different response due to the closure of stomata. The sentence should be rephrased to clarify this.

- p. 3, l. 26: The description of wind impact is misleading: High wind speed does indeed cause a stronger coupling, so it simply dampens the difference, both in case of Tl>Ta and in case of Tl<Ta. Hence, larger wind speed does not counteract the "benefits" of other variations, but it causes the responses to be dampened. The alleviating effect of these variations under high wind speed (so stronger coupling) is simply less, because of the overall smaller temperature difference.

---

## Referee Comment (RC2) · Anonymous Referee #2 · 9 May 2016

In their paper "Ideas and perspectives: Heat stress: more than hot air" De Boeck et al. emphasize the importance of leaf temperature rather than air temperature as the fundamental driver of heat stress in plants. Using an energy balance model along with field data they attempt to identify the drivers for differences between leaf and air temperature. Their ultimate goal is to educate ecologists and agronomists to improve their understanding of how heat waves can induce plant heat stress.

I agree with the authors that this is a relevant topic and that the importance of leaf temperature deserves more attention. However, I do not feel that the paper provides sufficient insights to actually inform scientists concerned with the analysis of heat waves. My concerns are:

1. The fundamental problem that the paper wishes to address is that many studies of

heat stress rely on air temperature rather than leaf temperature as a measure for heat stress, and therefore fail to reproduce or correctly attribute the impact of heat waves. While this is probably true, the use of just three references to underline this point is not particularly striking. In order to highlight the relevance of the issue, the paper should demonstrate that the use of air temperature is a common problem in ecological and agronomic studies across all scales, whether they analyze data or apply modeling. I don't think a thorough literature review is needed here. But a brief concise overview with examples from a wide range of applications is a minimum requirement.

2. The paper relies on an energy balance model described in another paper (De Boeck et al., 2012) to demonstrate the influence of various environmental variables on the difference between leaf and air temperature. The observed patterns are then discussed in the context of the physical processes that govern the heat and mass exchange in the soil-plant-atmosphere system. The authors thereby also resolve "counterintuitive" results such as the influence of wind speed and humidity. However, all this has been established textbook knowledge for nearly half a century, and there certainly isn't anything surprising or counterintuitive about it! The whole problem can basically be described by just four simple equations that not surprisingly are also used in the applied energy balance model (eqs. 1, 8, 9, and 12 in De Boeck et al., 2012). A thorough inspection of these fundamental equations rather than a superficial analysis of casually obtained results from the energy balance model (which essentially remains a black box to the reader), would be a far more educative exercise. For example, one could easily combine these four equations into something like $Tl = Ta + X - Y$ and demonstrate how the variables in X and Y determine whether leaf temperature is above or below air temperature. A series of contour plots of $Tl–Ta$ for different combinations of environmental variables and stomatal control could be used to quantify their relative importance and highlight important interactions. Such a more fundamental treatment of the issue would help the reader to develop a basic understanding of the physical laws determining leaf temperature, eventually stimulating the improvement of studies on plant heat stress.

[Figure]

3. Using some field data to demonstrate that theory holds true in practice is an excellent way to strengthen the argument of the paper. Unfortunately, the effect of only one variable is investigated although the data presumably would support a much wider range of relationships. The paper would gain a lot if the data were used to further explore the influence of other variable in the field. Additional important insights might be gauged from the analysis of diurnal variations.

4. Accurate analysis and proper use of statistical methods is crucial, even if the data is just used to illustrate a theoretical argument! Using the slight visual separation of data points obtained at days without irrigation in Figure 2 to support the argument that stomatal closure reduces transpiration cooling is farfetched, if not entirely wrong. While the statistical significance of the difference remains unknown, the attribution to stomatal closure simply has no basis. The difference could well be caused by slight variations in environmental conditions.

5. The description of the field experiment lacks detail. Information about the type and timing of irrigation is required to understand the potential influence by a wet canopy. Also, information on vegetation cover and the potential impact of bare soil on measurements is missing. At least the measurement principles and some basic specifics of "custom made" sensors should be mentioned. The rationale for mounting the radiation sensors unusually close to the surface and the potential impacts on measurements should be discussed.

---

## Short Comment (SC1) · 15 May 2016

An elaborate response will follow in the full revision where we will address the points raised by referee 1. Please note that the model was already validated (De Boeck et al. 2012, New Phytologist) on the same kind of vegetation as used in the short field test in the current study.
* * *

---

## Short Comment (SC2) · 15 May 2016

An elaborate response will follow in the full revision where we will address the points raised by referee 2. Two points of note:

- While we agree that physical processes governing heat exchange have been discussed in textbooks, we do not agree that this is established knowledge. Many heat wave and climate change studies still base hypotheses on heat stress and/or temperature effects on air temperatures rather than tissue temperatures. We felt that a concise manuscript clearly demonstrating the importance of other environmental variables on tissue temperatures would be very useful to many researchers (e.g. for improving their experimental set-ups). We will amend the manuscript to better introduce and discuss this.

[Figure]

- The field data resulted from a natural heat wave that occurred while we were setting up a field test of a new control for infrared heaters. Most sensors had been installed, but the wind sensor malfunctioned and had to be replaced later by another model, so that no data of wind speed were available during the short natural heat wave. This means that we cannot directly reconstruct the energy balance, limiting the amount of relationships we can empirically study. Still, we think that the data we do have, contribute to the manuscript. In the revised manuscript, we will address the referees' comments on the experimental part as best we can, although within the limitations posed by the set-up. The model is still the center piece of the study, and, as the earlier validation was convincing, should definitely bring out trends that the reader can trust.

―――――――――――――――――――――

---

## Author Comment (AC1) · 2 Jun 2016

**Response Letter**

We thank the referees for their constructive suggestions. We respond to each comment (italics) below.

The advice by the two referees helped us to make the study more comprehensive, while we preserve our initial idea of a crisp paper that is very clear about the major trends so that it is understandable and of practical use for a large audience (cf. comments referee 1). Extra analyses are presented as supplementary material, so that researchers looking for deeper understanding are not neglected. Moreover, we now specify that the applied model is free for use to maximise practicability. We could put it up on a site of Biogeosciences if possible, or alternatively, host it at another website. We revised the paper already in full at this stage, even though we cannot add it here yet. The most important additional figures have been inserted below the referee comment with which it is linked.

**Referee 1**

*De Boeck et al. provide a nice and concise manuscript discussing the importance of the use of leaf temperatures rather than air temperatures for addressing heat stress. The study applies a leaf temperature model published in De Boeck et al. (2012) for a set of sensitivity tests to address the importance of wind speed, relative humidity, radiation levels and leaf size for leaf temperatures. These sensitivities are used to discuss the variations in leaf temperature that can arise through meteorological conditions. In addition, leaf temperature measurements from a young grass stand are analysed to address the importance of drought for leaf temperatures.*

*The manuscript is nice and short, and although very concise, I consider the setup chosen here (discussion of the importance illustrated with an idealized set of sensitivity simulations), appropriate to emphasize the authors' opinion.*

→ In light of this, we tried to include the suggestions made by both referees without changing the set-up and format of the manuscript too much.

*However, the analysis of the leaf temperature measurements that are used to illustrate the drought impact need further attention - these can be used potentially to validate the model and to emphasize the conclusions drawn from the modelling, but in its current form they are not analysed in great depth, nor quantitatively compared with the modelling. I would recommend to use these to a greater extent (further comments below).*

*If the measurements can be integrated more in the rest of the manuscript, I expect this manuscript to be an attractive contribution to the discussion on analysis of heat stress and an important message for impact studies.*

*Major comments:*

*The data set of leaf temperature is an interesting contribution to the manuscript, but its analysis and the comparison with the modelling is too concise in its current form. I would recommend the following:*

*(1) a statistical analysis of the two sets (with and without irrigation) to determine whether the difference is significant;*

→ We included this analysis now (p 9 li 4-6), which shows that the difference between the two sets is significant indeed, and added some further clarification on p 4 li 21-22.

*(2) a validation of the model by using the measured energy fluxes and air temperature to simulate leaf temperature, which can be subsequently compared with the observations (if all model parameters are available or can be estimated);*

→ The model was already validated quite convincingly (De Boeck et al. 2012, New Phytologist) on the same type of vegetation (grass stand, p li 30-31). Regarding the field data used in the manuscript, we made opportunistic use of a set-up that was made ready for the testing of a new infrared heater control (cf. De Boeck & Nijs 2011, Journal of Ecology). Just prior to that test (which was carried out 8-24 July), there was a natural heat wave. Because of the homogeneous vegetation and because several sensors were already installed (apart from a wind sensor, which unfortunately malfunctioned just before the heat wave), the data from that heat wave period seemed appropriate to supplement our model results. However, because wind speed and stomatal conductance were not measured, the use of the heat wave data is restricted. This is why limited emphasis was put on these data, which should be seen as having a supporting role only, with the model (as validated earlier) forming the backbone of the current study. An extra analysis regarding the realism of the slope of the relationship between RH and $T_l$-$T_a$ was added (Fig. S8, and see explanation below).

[Figure]

**Figure S8:** Modelled influence of relative humidity (with 0.05 intervals) on leaf ($T_l$) – air ($T_a$) temperature differences. Input data reflect conditions similar to those in Fig. 2: $T_a$ = 30 °C, incident shortwave radiation was varied between 400 and 800 W m$^{-2}$, stomatal conductance = 0.4 μmol m$^{-2}$ s$^{-1}$, wind speed was varied between 0.5 and 0.8 m s$^{-1}$. Wind speed data at our site were unfortunately not measured during the heat wave due to sensor malfunction and were derived from data of a nearby meteorological station (Lint, Belgium) and correlation ($R^2$ = 0.80) with data registered on later days (9-23 July).

*(3) a derivation of the theoretical relationship between the temperature difference and RH using the model, possibly even for different cases (e.g., high/low wind speed), to determine whether the slope found by linear regression reflects the (range of the) theoretical behaviour;*

→ The requested model analysis was added as a supplementary figure (Fig. S8). The slope of that relationship and the values of $T_l$-$T_a$ are quite comparable to the ones that were measured in the field (p 4 li 19-20).

*(4) clarifying the figure caption of Fig. 2: Is the linear regression for the entire data set, or only for the irrigated days? If the two sets differ significantly, there could be separate regression lines for the two;*

→ This is now clarified (p 9 li 4). A regression line for the 'dry day' data was not added because that regression was not significant.

*(5) a discussion on the cause of the high scatter in the observed temperature differences: Are these measurement uncertainties, or can they be explained by the other variables not separated in the figure (stomatal conductance, wind speed).*

→ The scatter indeed results from co-varying factors (radiation, stomatal conductance, wind, cloud cover, atmospheric pressure), now mentioned on p 9 li 5-6, and can also be seen in Fig. S8 – even if co-variation was more limited there (e.g. $g_s$ and $T_a$ were fixed).

*It is striking to see that there may indeed be a smaller latent cooling for the non-irrigated day (this would need to be confirmed by statistical analysis - the spread is large), but that there is little impact on the range of observed temperature differences, so the response to other factors that cause the spread may not be affected that much.*

→ The limited amount of field data we have, render it difficult to make strong statements about this. The additional contour plots give more information on which variables could reduce or increase variation (e.g. low wind speed and low $g_s$ increase variation).

*Minor comments:*

*- p. 2, l. 17: the sensible heat flux is not mentioned in the discussion of the components of the energy balance.*

→ The sensible heat flux was already shortly mentioned on p 2 li 21-22. If additional explanation needs to be added, we will do so.

*- p. 3, l. 7: Please provide more information on the setup with the custom-made noncontact thermometer. I presume it measures infrared radiation? How do you ensure that you measure leaf temperature and not ground temperature? A five-week old and 10 cm high grassland will presumably have a rather low LAI (if LAI was measured, it would be great to have it reported in the manuscript of course).*

→ More details are now given (p 3 li 15-18), and a picture of the plot was added to show that the canopy cover was fairly high (Fig. S1). Some bare soil was likely still visible for the infrared temperature sensor, although its influence was likely limited. As the grass stand was still young and shallow-rooted, the top soil and vegetation would have been relatively coupled (as opposed to older vegetation which can draw water from deeper soil layers): when the top soil dries (and thus warms), such young plants would also suffer from water limitations and warm subsequently (cf. Fig. 2).

[Figure]

**Figure S1:** Homogeneous grass stand equipped with two pyranometers (upward and downward, for each hemisphere), a non-contact infrared thermometer for canopy temperature and a combined air temperature and relative air humidity sensor shielded by wooden panel.

*- p. 3, l. 15: "without extra energy" is somewhat misleading here: This is used to describe the low radiation case (100 W/m2), which of course does resemble a low level of solar energy. A case without extra energy (darkness) would have yet a different response due to the closure of stomata. The sentence should be rephrased to clarify this.*

→ The sentence has been rephrased for clarification (p 3 li 25). The additional figure S2 now also shows the effect of radiation along a range of stomatal conductance values (cf. nighttime stomatal closure).

*- p. 3, l. 26: The description of wind impact is misleading: High wind speed does indeed cause a stronger coupling, so it simply dampens the difference, both in case of Tl>Ta and in case of Tl<Ta. Hence, larger wind speed does not counteract the "benefits" of other variations, but it causes the responses to be dampened. The alleviating effect of these variations under high wind speed (so stronger coupling) is simply less, because of the overall smaller temperature difference.*

→ The starting point for our simulations is the 40 °C threshold (p 3 li 1), making any variable that leads to cooling (such as less radiation and higher $g_s$) 'beneficial' for plants in the short term as it helps them to avoid heat stress. When we write that higher wind speeds counteract such beneficial effects, we therefore do not think this is misleading. The explanation given by the referee – which is independent of heat stress thresholds - is also mentioned in the text as the mechanism behind the observations ("closer coupling between the plant and the air").

**Referee 2**

*In their paper "Ideas and perspectives: Heat stress: more than hot air" De Boeck et al. emphasize the importance of leaf temperature rather than air temperature as the fundamental driver of heat stress in plants. Using an energy balance model along with field data they attempt to identify the drivers for differences between leaf and air temperature. Their ultimate goal is to educate ecologists and agronomists to improve their understanding of how heat waves can induce plant heat stress.*

*I agree with the authors that this is a relevant topic and that the importance of leaf temperature deserves more attention. However, I do not feel that the paper provides sufficient insights to actually inform scientists concerned with the analysis of heat waves.*

*My concerns are:*

*1. The fundamental problem that the paper wishes to address is that many studies of heat stress rely on air temperature rather than leaf temperature as a measure for heat stress, and therefore fail to reproduce or correctly attribute the impact of heat waves. While this is probably true, the use of just three references to underline this point is not particularly striking. In order to highlight the relevance of the issue, the paper should demonstrate that the use of air temperature is a common problem in ecological and agronomic studies across all scales, whether they analyze data or apply modeling. I don't think a thorough literature review is needed here. But a brief concise overview with examples from a wide range of applications is a minimum requirement.*

→ We added two additional sentences highlighting that many studies on heat waves do not measure surface temperatures (p 2 li 5-8) - which is illustrative of the issue we address (if you do not measure these temperatures, you cannot take them into account properly) – and that also in modelling, air temperatures are often considered rather than canopy temperatures (p 2 li 8-10). We take a fairly careful approach, as we do not want to make other authors feel 'named and shamed'. We explicitly picked recent examples from leading journals in ecology and agronomy to support our argument.

*2. The paper relies on an energy balance model described in another paper (De Boeck et al., 2012) to demonstrate the influence of various environmental variables on the difference between leaf and air temperature. The observed patterns are then discussed in the context of the physical processes that govern the heat and mass exchange in the soil-plant-atmosphere system. The authors thereby also resolve "counterintuitive" results such as the influence of wind speed and humidity. However, all this has been established textbook knowledge for nearly half a century, and there certainly isn't anything surprising or counterintuitive about it!*

→ The referee is right that energy balance modelling is not new, and that the basics behind our study can be derived from textbooks. Nevertheless, the fact that many if not most studies on heat waves (and this is also true more in general for warming experiments) do not consider leaf/tissue temperatures supports the notion that a study such as this, presenting other important factors that govern tissue temperatures in an accessible and concise manner, would be of value for ecologists and agronomists alike. Moreover, textbooks do not contain a separate analysis focused on the leaf to air temperature patterns specifically for heat waves, which is in our opinion why its importance in those situations is often overlooked.

*The whole problem can basically be described by just four simple equations that not surprisingly are also used in the applied energy balance model (eqs. 1, 8, 9, and 12 in De Boeck et al., 2012). A thorough inspection of these fundamental equations rather than a superficial analysis of casually obtained results from the energy balance model (which essentially remains a black box to the reader), would be a far more educative exercise. For example, one could easily combine these four equations into something like Tl = Ta + X – Y and demonstrate how the variables in X and Y determine whether leaf temperature is above or below air temperature.*

→ The calculation of leaf temperatures in our model requires iterative computation and decision schemes (p 2 li 27-30 and De Boeck et al. 2012, New Phytologist, top of page 3 and Fig. 1), which is at odds with a straightforward unifying formula such as suggested. Our approach of showing what can be expected in opposite cases (low – high radiation, etc.) was meant to provide a clear picture of major trends also for non-specialists. The contour plots we now added as supplementary material should offer more detailed information for those readers that are looking for a more thorough understanding. Furthermore, we also put up the model as free to use (cf. p 2 li 31). This maximises practicability and gives all the workings of the model so that it no longer is a black box.

*A series of contour plots of Tl–Ta for different combinations of environmental variables and stomatal control could be used to quantify their relative importance and highlight important interactions. Such a more fundamental treatment of the issue would help the reader to develop a basic understanding of the physical laws determining leaf temperature, eventually stimulating the improvement of studies on plant heat stress.*

→ As per the suggestion, we added contour plots for all pairwise combinations of the major variables discussed in the manuscript (Fig. S2-7, example given below). To reconcile preserving the accessible nature of the study (as also fits the 'ideas and perspectives' format and the opinion of referee 1) with providing more detailed information, we added these plots together with further details and analyses in the supplementary material section (cf. p 3 li 5-6). As noted before, we now explicitly state that our model is free to use.

[Figure]

**Figure S7:** The influence of relative air humidity and wind speed on the difference between leaf ($T_l$) and air ($T_a$) temperatures (depicted by different colours). Generally, higher air humidity leads to relatively warmer leaves. Low wind speeds exacerbate effects of air humidity, while high wind speeds dampen these. Other variables were kept constant: air temperature = 40 °C, stomatal conductance = 0.2 mol m$^{-2}$ s$^{-1}$, incident shortwave radiation = 800 W m$^{-2}$ and leaf diameter = 0.005 m.

*3. Using some field data to demonstrate that theory holds true in practice is an excellent way to strengthen the argument of the paper. Unfortunately, the effect of only one variable is investigated although the data presumably would support a much wider range of relationships. The paper would gain a lot if the data were used to further explore the influence of other variable in the field. Additional important insights might be gauged from the analysis of diurnal variations.*

→ As stated in response to referee 1: The field data used in the manuscript are the result of the opportunistic use of a set-up that was made ready for the testing of a new infrared heater control (cf. De Boeck & Nijs, J Ecol 2011). Just prior to that test (which was carried out 8-24 July), there was a natural heat wave. Because of the homogeneous vegetation and because several sensors were already installed (apart from a wind sensor, which unfortunately malfunctioned just before the heat wave), the data from that heat wave period seemed appropriate to supplement our model results. However, because wind speed and stomatal conductance were not measured, the use of the heat wave data is restricted. This is why limited emphasis was put on these data, which should be seen as having a supporting role only, with the model forming the backbone of the current study.

*4. Accurate analysis and proper use of statistical methods is crucial, even if the data is just used to illustrate a theoretical argument! Using the slight visual separation of data points obtained at days without irrigation in Figure 2 to support the argument that stomatal closure reduces transpiration cooling is farfetched, if not entirely wrong. While the statistical significance of the difference remains unknown, the attribution to stomatal closure simply has no basis. The difference could well be caused by slight variations in environmental conditions.*

→ We performed a formal analysis in the revised manuscript, and this shows that both datasets differed significantly (p 9 li 4-6). The attribution to stomatal conductance is based on rational arguments: the grass stand was young and shallow-rooted, and the prevailing heat wave conditions led to high atmospheric water demand. One day without irrigation led to visual leaf wilting. Furthermore, wind speed (daily basis, derived from nearby meteorological station) was exactly the same for both datasets, and $R_{abs}$ was c. 15% lower on the dry day, which would have led to cooler rather than warmer leaves (Fig. S3, and note that the relationship between RH and radiation on $T_l$-$T_a$ is near-linear). We do agree that our assertion of lower $g_s$ is still speculative, which is why we toned down the wording further, and treat the difference between the datasets rather low key (p 4 li 20-23).

*5. The description of the field experiment lacks detail. Information about the type and timing of irrigation is required to understand the potential influence by a wet canopy. Also, information on vegetation cover and the potential impact of bare soil on measurements is missing. At least the measurement principles and some basic specifics of "custom made" sensors should be mentioned. The rationale for mounting the radiation sensors unusually close to the surface and the potential impacts on measurements should be discussed.*

→ More details have been added (p 3 li 15-18), as well as a picture of the plot (Fig. S1). The plot was set up to test a new infrared heater control (cf. De Boeck & Nijs, 2011, J Ecol), which is why radiometers needed to be mounted below the infrared heaters (cf. reply to comment 2 by referee 1) and relatively close to the canopy. This is also described in the paper of Kimball (Journal of Agronomy, 2015).

---

## Author Response (AR2)

Thank you for providing us with an opportunity to improve our manuscript further. We provided clarifications and edited a figure in line with the referees' suggestions. Referee 2 does not believe this manuscript is novel, which we do not agree with. As we explain in our responses, there is – to the best of our knowledge – no literature yet that explicitly examines effects on leaf or canopy temperatures during different-type heat waves. Our opinion on novelty is reinforced by the responses we received when presenting the results from this paper at ESA 2016 ("Plants and extreme events: Environmental buffers and amplifiers of heat extremes"), which were positive and surprised. We are therefore confident that the paper will be useful to many researchers, especially with the extensive background information presented. We suggest to make the model (Matlab code) available as free-to-use (the URL, if provided by Biogeosciences, can be filled in on p 2 li 32).

We respond to each comment (italics) below. Line numbers refer to the clean version of the document.

Sincerely,

Hans De Boeck & co-authors

**Referee 1:**

*Thank you for the revisions and the comprehensive answers to my earlier comments. I appreciate the changes made, in particular the contour plots that were added to the appendix.*

*The comparison between model and data has improved somewhat, but is still a weak point of the manuscript. Although the model has indeed been evaluated in De Boeck et al. (2012), the conditions in this earlier study were considerably more moderate (up to 20 C), and did not reflect heat stress. The relationship with relative humidity (Fig. 2) explains 13% of the variance, so there is a lot more to be explained by the other factors. Also, if relative humidity appears to be correlated with any of the other factors (which seems likely, e.g. when considering the diurnal pattern of several of these parameters), the relationship found in the regression may be flawed. This should at least be discussed.*

➔ Response: We are happy that our prior changes were well received by the referee. We agree that the model results are not explicitly validated for the high temperatures addressed in the manuscript. Apart from practical concerns (heat waves are rare and we would need multiple types, lab data are not an option since the environment is incomparable to outside), it is important to note that energy balance calculations have been used in hundreds of scientific papers and are widely accepted to yield good approximations of reality. Indeed, they are based on longstanding physical equations rather than on equations trying to grasp the complexity of biological processes. The solution of the equations for $T_l$ is part of many textbooks and at the heart of the environmental biophysics and meteorology disciplines. For this reason we do not agree that a lack of explicit comparisons between model and data is a weak point of the manuscript. We seem to have ended up between a rock and a hard place as Referee 1 would like to have some more certainty about the results, while Referee 2 considers them as obvious (which is opposite to uncertain). We now more explicitly mention the reliability of energy balance modelling on p 2 li 15, also in accordance with Referee 2's suggestion to refer to the relevant standard works.

The fact that the relationship with relative humidity (Fig. 2) explains only 13% of the variance was to be expected. Our simulations demonstrate that several other factors than RH also strongly determine $T_l$ at $T_a$= 40 °C, notably radiation, wind speed and stomatal conductance, and these all vary (as noted in the caption of Fig. 2). The relationship in Fig. 2 is not a validation of the model as it explicitly considers only one of the variables of the model, but rather an illustration of a general trend predicted by the model and also observed in reality. We state as such on p 4 li 16-17: the natural heat wave "provided us with an opportunity to test whether increasing air humidity diminishes the cooling capacity of leaves". The relevance here is that we indeed found a trend that was in line with the model's predictions regarding the effect of relative humidity on leaf cooling.

As the referee notes, RH indeed changes depending on other variables, notably air temperature. However, our simulations consider different situations at one temperature (40 °C), rather than the ramp up towards or the cooling down afterwards, exactly in order to obtain clean comparisons at the same peak air temperature. We agree that we did not explicitly mention the fact that the regression in Fig. 2 includes air temperatures before and after the peak, which is not the same as in the simulations with constant $T_a$. This has now been clarified (legend of Fig. 2).

*Fig. S8 gives a glimpse of other factors that could play a role, but it could differentiate the other factors varied better, e.g. by making separate regression lines, or by colouring the circles by SW radiation and/or wind speed.*

➔ Response: We edited the figure, making a visual distinction between the different radiation levels used in the simulation and explaining in the legend that "lower wind speed leads to lower $T_l$-$T_a$ at identical other environmental conditions". This should give the readers an improved idea of 'noise' in the relationship between relative humidity and $T_l$-$T_a$ caused by wind and radiation (cf. Figs. S2-S7).

*The contour plots are very nice and illustrative, I would suggest to extract the explanatory lines from each of the captions ("Generally, …") of S2-S7, and incorporate these in the main text (in some cases, these effects are already discussed there and can simply be removed from the captions).*

➔ Response: The problem is that variables here are compared in pairs, which is not the way the text in the manuscript is structured (figures relate to multiple paragraphs). We would therefore prefer to keep the explanations in the captions, where they directly relate to the figure in question. If the editor wishes to have this implemented in the main text nevertheless, we will of course do so.

**Referee 2:**

*The revised manuscript addresses some of the raised concerns but the use of an inappropriately complex model in favor of a thorough theoretical reasoning remains the fundamental flaw of the paper.*

➔ Here we disagree. If the model would be overly complex, it would not be part of many textbooks as the referee mentions below. Moreover, no part of the model can be omitted because all processes of energy exchange (sensible heat transport, latent heat transport, shortwave radiation fluxes and longwave radiation fluxes) have to be defined to be able to calculate leaf temperature. We also do not understand why using a model would constitute a fundamental flaw, nor can we see how the results in the figures (including their magnitudes for all combinations of factors) could be derived through theoretical reasoning.

*In their response, the authors justify the use of their model by claiming that the calculation of leaf temperature "requires iterative computation and decision schemes." This alone wouldn't justify the replacement of theory by numerical simulations. But it is not even true. An excellent example of a concise theoretical discussion of the topic including an explicit equation to determine leaf temperature from air temperature, radiation, wind speed, and vapor deficit can be found in Campbell and Norman (1998). It is not the only example. In fact, the issue is a standard topic covered by almost every textbook on the matter. Further examples include Geiger (1950), Oke (1987), Lambers, Chapin III, and Pons (2008), Monteith and Unsworth (2009), and Jones (2013), to name just a few.*

➔ Response: We agree that $T_l$ can also be calculated by an explicit equation (using approximation through linearization, as mentioned in Campbell & Norman 1998) instead of iterative computing, but the two are equivalent as both techniques calculate steady state by making the energy gains equal to the energy losses. We do therefore not understand how this would invalidate the results, especially as the referee does not claim that the iterative procedure would be wrong.

*Given the fact that knowledge about the difference between leaf and air temperature and its importance is so well established, it is indeed surprising that leaf temperature is not routinely used when assessing plant responses to heat waves and environmental conditions in general.*

➔ Response: This is exactly our point. Moreover, the emerging literature on extreme event ecology does not take into account that $T_l$ can be extremely different from $T_a$ during a heatwave, which is why we made the calculations at 40 °C. We presented the results discussed in the paper at the 2016 ESA meeting in Florida, and we received many positive and surprised responses from the ecologists present, which reinforces our view that – in spite of the established background knowledge on energy exchange – our simulations and discussion thereof are relevant and timely.

*One probable reason might be that leaf temperature is not a standard meteorological variable and cannot be easily measured on the landscape scale. On the other hand, estimation of leaf temperature from standard climate variables is data intensive and entails potentially large uncertainties. Additional problems arise from temperature gradients within canopies leading to non-uniform distribution of heat stress. Practical guidance on how to address these problems is required to lower the barrier for including leaf temperature in ecological and agronomic studies.*

➔ Response: Infrared imaging has now sufficiently advanced to make measurements of $T_l$ feasible both from a practical and financial point of view. We hope this paper will stimulate researchers to fill the gap and start making such measurements. Practical guidance is now added in a new paragraph on p 4 li 30 – p 5 li 5.

*In my view, the manuscript by De Boeck et al. fails to provide any new ideas and perspectives on the issue of leaf temperature and heat stress. It merely states that using air temperature instead of leaf temperature as a measure for heat stress is inappropriate. The attempt to identify the drivers for leaf temperature remains superficial and consequently ignores the vast body of literature on the topic. The paper does not provide any practical suggestions for improving studies on heat waves nor does it line out a way towards addressing the problem more appropriately in the future.*

➔ Response: We do not understand why the characterisation of $T_l$ in many different heatwave conditions would be superficial. To our knowledge there is no body on leaf temperature during extreme temperature events (cf. p 1 li 10-11), and the referee does not list any existing publications on the subject. We believe that our assessment is important and novel with regards to extreme event ecology, as also emerges from contacts with colleagues (see above). We do agree that we failed to implement practical suggestions and ways forward. This has now been amended (p 4 li 30 – p 5 li 5).

---

## Author Response (AR3)

**Referee comments (marked in italics)**

*I would like to thank the authors for the revision of their manuscript. I understand their struggle with two opposing sets of comments, and think that the current version of the manuscript is a well-balanced compromise between these. I have a few minor suggestions (below), and I recommend publication of this manuscript to support the discussion and usage of leaf temperature (measurements and models) in our analysis of heat wave impacts.*

→ Response: We are grateful for the appreciation of our work and the revision.

*- p. 1/l. 9: replace "Climate change models" with "Climate models"*

→ Response: changed as suggested (p 1 li 9)

*- p. 1/l. 15: the sentence "in excess of 10C … to even 20C" could be misunderstood: The range is not between 10 and 20 degrees, but rather between 0 and 10/20. Also, the given upper limits (10/20 degrees) are obtained only when taking rather extreme (and possibly very unlikely) conditions (such as high RH and closed stomata, Fig. S9) into the comparisons. I do not think that this is within the scope of something the user would understand as a "fluctuation" (p. 1/l. 15), the conditions that need to be met to obtain these differences are so extreme that they will rarely occur within a short period of time. I would recommend a more moderate statement that better reflects real conditions.*

→ Response: We rewrote part of the sentence in a neutral tone (p 1 li 14-15). This should avoid further misunderstandings.

*- p. 3/l. 12: The addition "via stomatal responses" is unnecessary here, the authors cannot exclude other impacts (e.g. a change in near-surface RH resulting from the change in irrigation).*

→ Response: omitted as suggested (p 3 li 12)

*- p. 4/l. 30: replace "such as used here" with "as used here", or "such as the one used here".*

→ Response: changed as suggested (p 4 li 30).

*- p. 4/l. 30: I do not think that model and IR observations exclude each other: We will need more IR observations to evaluate model performance (e.g. to perform a more direct comparison under heat wave conditions). But in general I like the addition of the statement on availability/possibility of IR measurements.*

→ Response: We rewrote the sentence to avoid the idea that both are mutually exclusive (p 4 li 30-32).